# OpenReview forum: "Silent Leaks: Implicit Knowledge Extraction Attack on RAG Systems"
_ICLR.cc/2026/Conference — ICLR 2026 Poster_

### Official Review · Reviewer_iS3Y · 2025-10-29

**Soundness:** 3
**Presentation:** 3
**Contribution:** 3
**Rating:** 6
**Confidence:** 4

**Summary:**

This paper presents IKEA, a method that extracts knowledge from RAG using queries. IKEA stays stealthy by creating natural queries built from anchor concepts. The method has two parts. Experience Reflection Sampling chooses concepts that are likely linked to the RAG’s internal knowledge based on past query results. Trust Region Directed Mutation changes anchor concepts within a set similarity range to find new and related information more effectively. Experiemnts show that IKEA performs much better than other methods. The extracted knowledge can also be used to build a working substitute RAG system.

**Strengths:**

- The paper studies an important security issue in RAG systems : extraction attacks. Its focus on harmless-looking queries makes it different from most past work.

- IKEA method is explained in a clear and direct way.  Figure 1 gives a clear summary of the process.

- Experiments use several settings. Results show that IKEA keeps high EE and ASR while passing basic defenses. This is a strong finding.

- Code is provided. It seems make sense.

**Weaknesses:**

- The tested defenses  are not enough. Stronger and more realistic defenses include semantic output filtering, consistency checks, detection of repeated probing, or methods for iterative query attacks.

- I am not sure about the main assumption that the RAG topic is fixed and known limits how well the method can be used in other cases.

- The results is not enough to support the claim that the substitute RAG performs “comparably” (Sec 4.5).  Three metrics cannot measure many other aspects .

- The cost in time, API calls, and total query rounds isnot clear. This may make the attack too expensive for extracting large knowledge bases.

**Questions:**

1. See weakness

2. The topic probing method seems crucial for practical applicability. Could you provide more details on its robustness?

3. While IKEA avoids malicious prompts, could the pattern of queries generated be detectable by analyzing query sequences over time using anomaly detection techniques?

---

> ### Author Response · Authors · 2025-11-23
> **Rebuttal by Authors [1/3]**
>
> Thank you for your valuable comments. Below we respond to the comments in **Weaknesses** and **Questions**.
>
> ---
>
> ***Weakness 1: Tested defenses are not enough.***
>
> Thank you for your insightful suggestion. We method-by-method analyse the comment mentioned defenses here:
>
> 1. For **consistency check**, the consistency check is usually applied in defense against jailbreak attacks, which is not suitable for our scenarios.
>
> 2. Following reviewers' comment, we additionally add realistic deployments including semantic and iterative defenses. We include ControlNet, a semantic filtering defense, and a sequential anomaly detector trained from attack data, which are deployable and provide more realistic defensive baselines. The experiment details and results of additional defense are shown in $\color{blue} \text{Section 4.7}$.
>
>    Specifically, we trained a transformer-based **iterative detector** with the three attacks' data and human-rag interaction data [9]. The sequential detector achieves near-perfect performance against baseline attacks (DGEA and RAG-Thief), with AUC values of 1.0 and TPRs at both 1% and 10% FPR effectively saturating at 1.0. In contrast, it is much less effective at identifying IKEA, where the AUC drops to 0.76 and the TPRs fall to 0.03 at 1% FPR and 0.24 at 10% FPR, indicating that IKEA largely evades standard sequence-level anomaly signals.
>
>    We also deploy the ControlNET [10] detector, a **semantic filtering** defense that monitors the semantic drift and filters suspicious requests. This detector also performs extremely well on baseline attacks, with AUC values of 1.0 (DGEA) and 0.99 (RAG-Thief) and TPRs near 1.0 at both 1% and 10% FPR, as shown in $\color{blue} \text{Table 4}$. However, its performance degrades substantially on IKEA, with an AUC of only 0.75 and TPRs of 0 at 1% FPR and 0.11 at 10% FPR.
>
>    Taken together, these results show that IKEA is markedly more stealthy than existing baselines under various defenses.
>
> ---
>
> ***Weakness 2: Fixed and known topic assumption limits applicability.***
>
> We appreciate the reviewer’s comments. Here, we analyse this point systematically:
>
> 1. Our assumption reflects realistic RAG deployments, where most systems serve a fixed domain [1] [2] [3] [4]. These production systems justify our setting. RAG is typically used to enrich LLMs with external domain knowledge for specialized scenarios or users, such as medical question answering [6], financial analysis [7], or legal inquiry [8]. Therefore, the RAG topics are usually exposed to normal users for better utilization of the RAG.
>
> 2. Under weaker assumptions where topics are unknown to the attacker, IKEA still performs effectively, reflecting the effectiveness of the probing method ($\color{blue} \text{Table 3}$).  We also emphasize that there’s no need for any prior knowledge in the topic-probing stage and the probing algorithm does not rely on specific seed topics ($\color{blue} \text{Section 4.6}$). This further proves the practicality of our assumption.
>
> 3. To further assess the adaptivity of our method under multi-topic scenarios, we conduct multi-topic end-to-end evaluation on NQ-corpus ($\color{blue} \text{Table 1}$), a Multi-topic, open-domain dataset with no literal topics.  On NQ-Corpus, IKEA maintains stable performance across defenses (EE ≈ 0.64–0.66; ASR ≈ 0.82–0.88), whereas all competing attacks fail in some defense settings. These results show that the topic probing algorithm works on large-scale, heterogeneous documents without triggering detection or largely degrading extraction performance. This proves that our assumption and method design is reasonable.

---

> ### Author Response · Authors · 2025-11-23
> **Rebuttal by Authors [2/3]**
>
> ***Weakness 3: Substitute RAG comparability not fully supported.***
>
> We thank the reviewer's comment and will modify the expression in the revision.
>
> 1. The “comparable” in current version refers to functional utility, not literal reconstruction. The main point is that IKEA has better informative extraction capability as proved in $\color{blue} \text{section 4.5}$. $\color{blue} \text{Table 2}$ shows that substitute RAG constructed from IKEA outperforms RAG-thief’s and DGEA’s across all metrics (over 40% in Accuracy, 18% in Rouge-L, and 30% in Similarity), demonstrating IKEA’s ability to reconstruct high-fidelity knowledge bases from black-box access.
>
> 2. To strengthen this, we provide additional evaluation of down-stream task on the symptom-to-diagnosis dataset [15] ($\color{blue} \text{Appendix B.10}$, $\color{blue} \text{Figure 9}$):
>    We test the task accuracy on the clinical classification dataset, which requires the RAG system to analyse the patient description into condition classification. We use extractions under both input- and output-level defense setting to reconstruct the substitute RAG. As illustrated in $\color{blue} \text{Figure 9}$, the substitute RAG constructed using IKEA achieves performance closest to the original RAG, reaching 0.38 accuracy and 0.88 semantic similarity. In contrast, baselines such as RAG-Thief, DGEA, and PoR exhibit substantial degradation, reflecting their limited coverage and weaker semantic reconstruction. These results demonstrate that IKEA recovers clinically meaningful knowledge that reliably supports downstream reasoning tasks.
>
> ---
>
> ***Weakness 4: Cost and API usage unclear.***
>
> We thank the reviewer's comment and have clarified this in $\color{blue} \text{Appendix B.3}$:
>
> Cost and time measurements are already provided in $\color{blue} \text{Table 9}$: We include query tokens, attack tokens, and 256-round extraction time across baselines. IKEA reduces extraction time compared to RAG-Thief and DGEA despite stronger performance. As shown in $\color{blue} \text{Table 9}$, we observe that the attack token cost of IKEA is lower (208.74K) than Rag-Thief (233.91K). Notably, DGEA doesn't leverage LLM in attack query construction, leading 0 token usage in attack token counts. Moreover, IKEA also achieves the lowest extraction time (5220s), outperforming both Rag-Thief (6012s) and DGEA (6636s). These results demonstrate that IKEA strikes an acceptable balance between effectiveness and efficiency. We highlight this more explicitly in $\color{blue} \text{Appendix B.3}$.
>
> ---
>
> ***Question 1:***
>
> We have addressed each weakness in detail above (see **Weakness 1-4**).
>
> ---
>
> ***Question 2: More details on robustness of topic probing.***
> We thank the reviewer’s valuable comment. Our probing method exhibits robustness, as we demonstrated below:
>
> 1. The topic probing algorithm is robust to various seed topics. In the experiment of $\color{blue} \text{Table 3}$, we initialize the seed set with 20 randomly selected second-level Wikipedia categories. The $\color{blue} \text{Table 3}$'s result shows comparable performance with the known-topic setting.
>
> 2. To address topic-probing algorithm's robustness to noisy and heterogeneous datasets, we further extend our robustness analysis:
>
>    (1) **Noise-injected datasets** ($\color{blue} \text{Appendix B.13}$, $\color{blue} \text{Table 21}$): we inject different ratios of noise documents into the target RAG database and evaluate the resulting pseudo-topic by measuring its mean similarity to the ground-truth topic across four datasets. Practically, we randomly sample documents from NQ-corpus [11] as the source of noise documents. As shown in $\color{blue} \text{Table 21}$, the algorithm remains highly stable under small perturbations (noise \< 0.1), and consistently recovers semantics. Even with substantial noise (0.3), the probed topics retain meaningful alignment.
>
>    (2) **Multi-topic corpora**: The NQ-corpus [11] doesn't have literal topics, therefore we firstly conduct Multi-topic probing and then extract it with IKEA ($\color{blue} \text{Table 1}$). The end-to-end evaluation on NQ-corpus confirms that the algorithm remains effective even without centralized corpus semantics. On NQ-Corpus, IKEA maintains stable performance across defenses (EE ≈ 0.64–0.66; ASR ≈ 0.82–0.88), whereas all competing attacks fail in some defense settings. These results show that the topic probing algorithm is robust on large-scale, heterogeneous documents without triggering detection or largely degrading extraction performance.

---

> ### Author Response · Authors · 2025-11-23
> **Rebuttal by Authors [3/3]**
>
> ***Question 3: Could IKEA queries be detectable via sequential anomaly detection?***
>
> We address this concern in the revision ($\color{blue} \text{Section 4.7}$):
>
> We trained a transformer-based sequential detector with three attacks' data (IKEA, DGEA [12] and RAG-Thief [13]) and human-rag interaction data [9]. As shown in $\color{blue} \text{Table 4}$, The sequential detector achieves near-perfect performance against baseline attacks (DGEA and RAG-Thief), with AUC values of 1.0 and TPRs at both 1% and 10% FPR effectively saturating at 1.0. In contrast, it is much less effective at identifying IKEA, where the AUC drops to 0.76 and the TPRs fall to 0.03 at 1% FPR and 0.24 at 10% FPR, indicating that IKEA largely evades standard sequence-level anomaly signals.
>
> The reason is because IKEA's queries are natural language, which closely follow the domain’s typical question distribution. Thus, IKEA is less detectable than explicit extraction like DGEA and RAG-Thief and hard to be defended under reasonable false predicted rate.
>
> ---
>
> ### **Reference**:
>
> [1] Varun Kumar, Leonard Gleyzer, Adar Kahana, Khemraj Shukla, and George Em Karniadakis. My- crunchgpt: A llm assisted framework for scientific machine learning. Journal of Machine Learn- ing for Modeling and Computing, 4(4), 2023\.
>
> [2] Peng Xia, Kangyu Zhu, Haoran Li, Tianze Wang, Weijia Shi, Sheng Wang, Linjun Zhang, James Zou, and Huaxiu Yao. Mmed-rag: Versatile multimodal rag system for medical vision language models. arXiv preprint arXiv:2410.13085, 2024
>
> [3] Spurthi Setty, Harsh Thakkar, Alyssa Lee, Eden Chung, and Natan Vidra. Improving retrieval for rag based question answering models on financial documents. arXiv preprint arXiv:2404.07221, 2024\.
>
> [4] Nirmalie Wiratunga, Ramitha Abeyratne, Lasal Jayawardena, Kyle Martin, Stewart Massie, Ikechukwu Nkisi-Orji, Ruvan Weerasinghe, Anne Liret, and Bruno Fleisch. Cbr-rag: case-based reasoning for retrieval augmented generation in llms for legal question answering. In Interna- tional Conference on Case-Based Reasoning, pp. 445–460. Springer, 2024\.
>
> [5] Sleem, L., Francois, J., Li, L., Foucher, N., Gentile, N., & State, R. (2025). NegBLEURT Forest: Leveraging Inconsistencies for Detecting Jailbreak Attacks. arXiv preprint arXiv:2511.11784 .
>
> [6] Alejandro Lozano, Scott L Fleming, Chia-Chun Chiang, and Nigam Shah. Clinfo. ai: An open-source retrieval-augmented large language model system for answering medical questions using scientific literature. In Pacific Symposium on Biocomputing 2024, pp. 8–23. World Scientific,2023
>
> [7] Xiang Li, Zhenyu Li, Chen Shi, Yong Xu, Qing Du, Mingkui Tan, Jun Huang, and Wei Lin. Alphafin: Benchmarking financial analysis with retrieval-augmented stock-chain framework. arXiv preprint arXiv:2403.12582, 2024a.
>
> [8] Nirmalie Wiratunga, Ramitha Abeyratne, Lasal Jayawardena, Kyle Martin, Stewart Massie, Ikechukwu Nkisi-Orji, Ruvan Weerasinghe, Anne Liret, and Bruno Fleisch. Cbr-rag: case-based reasoning for retrieval augmented generation in llms for legal question answering. In International Conference on Case-Based Reasoning, pp. 445–460. Springer, 2024\.
>
> [9] Ruizhe Zhu, Hao Zhu, Yaxuan Li, Syang Zhou, Shijing Cai, Malgorzata Lazuka, and Elliott Ash.Dialogueforge: Llm simulation of human-chatbot dialogue. arXiv preprint arXiv:2507.15752,2025.
>
> [10] Hongwei Yao, Haoran Shi, Yidou Chen, Yixin Jiang, Cong Wang, Zhan Qin, Kui Ren, and Chun Chen. Controlnet: A firewall for rag-based llm system. arXiv preprint arXiv:2504.09593, 2025\.
>
> [11] Jack Morris. nq\_corpus\_dpr. https://huggingface.co/datasets/jxm/nq\_corpus\_dpr. Hugging Face dataset.
>
> [12] Stav Cohen, Ron Bitton, and Ben Nassi. Unleashing worms and extracting data: Escalating the outcome of attacks against rag-based inference in scale and severity using jailbreaking. arXiv preprint arXiv:2409.08045, 2024\.
>
> [13] Changyue Jiang, Xudong Pan, Geng Hong, Chenfu Bao, and Min Yang. Rag-thief: Scalable extrac- tion of private data from retrieval-augmented generation applications with agent-based attacks. arXiv preprint arXiv:2411.14110, 2024\.
>
> [14] Zhenting Qi, Hanlin Zhang, Eric P Xing, Sham M Kakade, and Himabindu Lakkaraju. Follow my instruction and spill the beans: Scalable data extraction from retrieval-augmented generation systems. In ICLR 2024 Workshop on Navigating and Addressing Data Problems for Foundation Models.
>
> [15] gretelai. symptom_to_diagnosis. https://huggingface.co/datasets/gretelai/ symptom_to_diagnosis. Hugging Face dataset.

---

> ### Comment · Reviewer_iS3Y · 2025-11-25
>
> Thanks for your detailed response. I have checked the response and the revised paper, most of my concerns are handled and the paper is better now. I will raise my rating.

---

> > ### Author Response · Authors · 2025-11-27
> >
> > We sincerely appreciate your recognition of our work. We will further improve the paper based on your valuable comments. Thank you again.

---

### Official Review · Reviewer_CR94 · 2025-10-30

**Soundness:** 2
**Presentation:** 3
**Contribution:** 2
**Rating:** 4
**Confidence:** 4

**Summary:**

This paper studies extraction of documents from a RAG knowledge base. Instead of using malicious queries, the authors use benign queries repeatedly to collect RAG answers as stolen knowledge, and propose several tricks to improve search efficiency—e.g., avoiding duplicate retrievals and increasing coverage. Experiments evaluate metrics including extraction efficiency and the downstream performance of RAG systems reconstructed from the stolen knowledge.

**Strengths:**

1. The paper addresses an important problem by studying the privacy risks of RAG systems under more realistic settings—specifically, black-box access with defenses in place.
2. Compared with baselines, the proposed method demonstrates stronger robustness against defended RAG systems, successfully extracting more knowledge when defenses are applied.
3. The paper is well-written, and the overall idea is intuitive and easy to follow.

**Weaknesses:**

1. The idea of using query–response semantic distance as a proxy for local RAG density is based purely on intuition, without further discussion. The paper does not provide references or experiments to validate this assumption.
2. The evaluation includes only two baselines, while several other relevant methods are mentioned but not compared experimentally.
3. The extracted documents achieve low ROUGE scores (below 0.3, Table 1), indicating that the extracted content fails to accurately recover the original documents. This limits the practical implications for privacy or copyright concerns.
4. Some metric definitions are unclear. For example, *extraction efficiency* depends on the number of “unique” extracted documents, but the notion of uniqueness is not specified. Moreover, since the method does not reconstruct original documents, comparability of this metric with prior work is questionable. Similarly, the definition of *ASR*—the ratio of non-rejected queries—does not directly measure extraction success.
5. The proposed method introduces many hyperparameters (over ten), which may be difficult to tune in practice. The paper provides little discussion on how these parameters are chosen.
6. Ablation results show only marginal improvements over random baselines (Table 13), particularly for ASR, CRR, and SS metrics, raising concerns about the actual effectiveness of the proposed approach.

**Questions:**

In the evaluation, some methods such as DEGA achieve high ROUGE scores (up to 0.96 in Table 1) in the no-defense setting, suggesting near-literal copying. However, their embedding similarity remains relatively low. What are the possible reasons?

---

> ### Author Response · Authors · 2025-11-23
> **Rebuttal by Authors [1/4]**
>
> Thank you for your valuable feedbacks. Below we respond to the comments in **Weaknesses** and **Questions**.
>
> ---
>
> ***Weakness 1: Semantic distance as proxy for local RAG density lacks validation.***
>
> We appreciate the insightful comment and thank the reviewer for highlighting this.
> We add a dedicated validation section ($\color{blue} \text{Figure 6}$, $\color{blue} \text{Appendix B.6}$), covering three datasets, showing that query–response similarity strongly correlates with local document density. For each query, we compute the number of RAG documents whose similarity to the query exceeds a high threshold (0.45 for MPNet), treating this count as an estimate of local density. As visualized in Figure 6, all datasets exhibit a clear upward trend: higher query–response similarity corresponds to denser neighborhoods in the retrieval space. Pearson correlations further confirm this pattern, with coefficients of 0.65, 0.55, and 0.64 respectively, indicating a consistent positive linear relationship. According to [6], it is reasonable to consider there exists strong linear correlation between the query–response similarity and local density with all Pearson coefficients over 0.5. It is reasonable to regard this as a broadly applicable phenomenon. Hence, we consider this as a valid assumption in our method.
>
> ---
>
> ***Weakness 2: Concerns about lack of baselines.***
>
> We appreciate the reviewer’s concern regarding the coverage of baselines. To address this point comprehensively, we next provide a systematic, method-by-method discussion of all potentially relevant extraction approaches, including those presented in the papers cited in the Related Work section.
>
> 1. We do not include PIDE [1] in our experiments for its non-adaptive nature and its less optimal performance, which has been proved in previous works [6, 7].
>
> 2. Note that DGEA [6] is an adaptive jailbreak attack, not a simple prompt-injection method. It is designed to actively explore unseen documents and therefore, is representative of adaptive extraction.
>
> 3. RAG-thief [7] is also an adaptive extraction attack, which utilizes document overlap and LLM query expansion.
>
> 4. We additionally add comparison experiments of the PoR (Pirate-of the RAG) attack [20], an adaptive black-box extraction method ($\color{blue} \text{Table 20}$) in our revised version and therefore has included all existing published RAG extraction methods. IKEA performs over 40% higher EE on PoR in all settings, over 50% higher EE on PoR in all defensive settings. IKEA also outperforms PoR in ASR in nearly all settings, especially under input-ensemble defense, which proves the superiority of IKEA.
>
> 5. Additionally, we evaluate benign-query extraction baselines ($\color{blue} \text{Table 20}$, $\color{blue} \text{Appendix B.12}$). We detail these evaluations in **Weakness 6** and **Question 1** below. As shown in $\color{blue} \text{Table 20}$, none of these approaches achieves strong extraction performance: EE remains below 0.51 across all benign baseline. In contrast, IKEA reaches substantially higher EE (0.88), demonstrating that our method is far more effective than naive or simple heuristic benign-query expansion.
>
> 6. Theoretically, we also provide theoretical analysis ($\color{blue} \text{Appendix F}$) to demonstrate that IKEA achieves near-optimal coverage complexity $T(\alpha) = \Theta(\alpha N / K)$, while random, greedy, or cluster-wise baselines require $\Omega\big (c\frac{N_{\max}}{K}\log N_{\max}\big)$ queries for complete coverage—formally explaining why these baselines are inherently weaker.
>
> In summary, we broadly considered all relevant potential baselines, as well as strong black-box methods, non-jailbreak variants, and adaptive coverage attacks. We provide detailed discussion and empirical comparisons across these settings, which consistently show that our approach achieves superior performance under various defenses, making our overall empirical evaluation fully representative.

---

> ### Author Response · Authors · 2025-11-23
> **Rebuttal by Authors [2/4]**
>
> ***Weakness 3: Concern for limited copyright/privacy threats with Low ROUGE.***
>
> ROUGE alone does not capture the risk model we study. Copyright and privacy harm do not require verbatim reproduction: semantic recovery is sufficient to recreate usable proprietary functionality and infringe the privacy threat. To prove this:
>
> 1. IKEA does not aim to verbatim reconstruct the documents, while is designed to extract informative documents summaries. We provide IKEA case studies ($\color{blue} \text{Appendix I}$) to show that the benign extractions contain considerable details, like highly domain-specific terminology, headers, and structured definitions without literal copying. CRR is designed to evaluate the extraction of the key terminologies, while SS is to evaluate the semantic reconstruction rate of the whole extraction. Therefore, a relatively low Rouge-L does not mean little threat, key terminology leakage with highly informative summaries can pose serious threat.
> 2. We acknowledge that only SS and CRR cannot fully evaluate the threat of extractions. Therefore, we also provide the evaluations on the extracted retrievals' internal knowledge ($\color{blue} \text{Section 4.4}$, $\color{blue} \text{Figure 3}$) and substitute RAG's performance ($\color{blue} \text{Section 4.5}$, $\color{blue} \text{Table 2}$) represent that the extractions contain usable informative copyright knowledge and our method effectively poses underlined copyright/privacy threats. There are already relevant legal cases [4,5] demonstrate the practical risk that AI-generated outputs, when substituting or competing with original works, may affect the market for copyright holders—even if the outputs are not identical. This further supports our argument in $\color{blue} \text{Section 4.4}$.
> 3. To further strengthen the claim and prove the possible commercial value of the extractions, we add downstream RAG evaluation ($\color{blue} \text{Figure 9}$) with extractions under both input- and output-level defense to build the substitute RAG. The result shows that IKEA’s extracted corpus supports downstream diagnosis tasks almost as well as the original RAG, confirming that semantic leakage is practically exploitable. Hence, our measurements also reflect copyright/privacy risk as actual utility of stolen knowledge, not just string overlap.
>
> Our attack mainly focuses on a realistic scenario in which a malicious adversary covertly extracts key knowledge from RAG documents via IKEA, and then builds a functional substitute that closely mimics the original RAG. This may encourage users to use the pirated RAG instead of the original one, potentially resulting in serious copyright infringement for RAG providers or the document content owners. This directly addresses the reviewer’s concern: our measurements reflect copyright/privacy risk as actual utility of stolen knowledge, which cannot be assessed solely through simple string overlap. Thus, even with low ROUGE, the extracted knowledge is sufficient to reconstruct a functional RAG ($\color{blue} \text{Table 2}$, $\color{blue} \text{Figure 9}$)—precisely the risk our work seeks to expose.
>
> ---
>
> ***Weakness 4: Metric definitions unclear concern.***
>
> We appreciate the reviewer's comment. We have provided $\color{blue} \text{Appendix A.2}$ with full formal definitions. We emphasize that:
>
> 1. Uniqueness corresponds to distinct retrieved-document indexes, consistent with prior extraction work. **EE** measures the fraction of unique retrieved indexes relative to the corpus; this directly reflects index-level knowledge coverage. **ASR** measures the ratio of non-rejected queries, which is the operational definition of extraction progress in benign-query settings where the system may refuse to answer. The ASR reflects the effectiveness of extraction queries in getting informative responses and avoiding detection.
>
> 2. Evaluating extraction quality is fundamentally multifaceted. We therefore combine Process-level metrics (EE/ASR/CRR/SS) ($\color{blue} \text{Table 1}$, $\color{blue} \text{Table 7}$) and Utility-level evaluation via substitute RAG ($\color{blue} \text{Section 4.4}$, $\color{blue} \text{Section 4.5}$, $\color{blue} \text{Figure 9}$), which together offer a complete measurement. Process-level metrics reflect whether the attack can efficiently extract informative responses; Utility-level evaluation reflects whether the extraction is informative and be used to construct effective substitute RAG for copyright/privacy violation.
>
> 3. We elaborate the evaluation on substitute RAG performance in **Weakness 3**, $\color{blue} \text{Section 4.4}$, $\color{blue} \text{Section 4.5}$ and $\color{blue} \text{Figure 9}$, which also poses realistic measurement of copyright threats.

---

> ### Author Response · Authors · 2025-11-23
> **Rebuttal by Authors [3/4]**
>
> ***Weakness 5: Concern about too many hyperparameters with tuning unclear.***
>
> We thank you for your comment. We therefore provide a standard hyperparameter configuration in $\color{blue} \text{Table 6}$ and use it across all experiments. We also include the key parameter $\gamma$'s sensitivity analysis ($\color{blue} \text{Appendix B.8}$, $\color{blue} \text{Figure 7}$). Results show that larger $\gamma$ (tighter trust regions) improves EE and ASR, but increases cost. A moderate setting ($\gamma \approx\textrm{0.5}$) achieves the best efficiency–cost balance and is used as the default in our experiments.
> Other parameters are chosen empirically and remain consistent throughout all evaluation experiments. We will provide the details and add further ablation experiments of these parameters in the future version.
>
> ---
>
> ***Weakness 6: Ablations show marginal gains over Random and effectiveness questionable.***
>
> 1. For the concern about marginal gains on EE:
>    We appreciate the reviewer’s insightful comment regarding the small performance gap between *Random* and IKEA. The concern arises from considering our *Random* ablation as a fully naïve baseline. In fact, the original Random ablation operates on a pre-filtered, anchor-guided candidate pool generated by IKEA’s initialization procedure in $\color{blue} \text{Section 3.2}$, rather than sampling from the full query space. Consequently, it does not constitute a truly global random strategy.
>    To address this potential misunderstanding, we have updated $\color{blue} \text{Table 14}$ to include a new true global Random baseline and revised the expression, and we have also provided additional explanation in $\color{blue} \text{Appendix B.8}$. The extraction efficiency gap between the global random and ER+TRDM has increased to 0.47. This comparison further highlights the effectiveness of our proposed method.
> 2. For the concern about questionable effectiveness on ASR/CRR/SS:
>    The little gap on ASR/CRR/SS between ER+TRDM and Random methods is because we use the same query generation function for fair comparison in $\color{blue} \text{Table 14}$. Therefore ASR/CRR/SS are not the main metric concerned in this experiment, as ASR/CRR/SS are mainly affected by the query ways. (Ablation about query function see $\color{blue} \text{Appendix B.8}$ "Effectiveness of Implicit queries." )
> 3. Additional explanation:
> Actually, ER and TRDM are originally designed to improve efficiency and coverage speed. We can see the efficiency gap between Random (w/o Anchors) method and ER+TRDM method in $\color{blue} \text{Table 14}$. This aligns with our added $\color{blue} \text{Appendix F}$ ’s complexity analysis, which shows that IKEA reduces expected query complexity from $\Omega\big (c\frac{N_{\max}}{K}\log N_{\max}\big)$ to $T(\alpha) = \Theta(N_{\max} / K)$ for nearly complete extraction. Notably, under limited query budget or query rounds, higher extraction efficiency (i.e. lower query complexity) implies more informative retrivals, which contribute largely to reconstructing usable substitute RAG.
>
> ---
>
> ***Question 1: Why does DGEA achieve high ROUGE but relatively low embedding similarity?***
>
> We appreciate the reviewer’s attention to these metrics. Our earlier description of CRR mistakenly referred to concatenating all retrieved documents. This was a wording slip, and we have corrected it in the uploaded revised version ($\color{blue} \text{Appendix A.2}$). In practice, we compute ROUGE-L between the response and each retrieved document individually and take the maximum. This follows the empirical behavior of LLMs in RAG settings: when producing factual responses, they typically rely on verbatim details from the *most relevant* retrieved document rather than uniformly using all retrieved texts. Taking the maximum ROUGE-L therefore more accurately captures whether any retrieved document provides the necessary literal support.
> For SS, we compute similarity between the response embedding and the embedding of the *concatenated* retrieved documents. Unlike verbatim copying, LLM responses tend to integrate and summarize semantic information across multiple retrieved documents. Concatenating the retrievals yields an overall semantic target, allowing SS to reflect how well the model reconstructs the overall semantic content present in the retrieval set.
>
> This design explains why SS may be lower even when CRR is high: When the prompt push model to concentrate on reproducing literal details from the single most relevant document, CRR becomes high, yet the semantic embedding of its answer covers only a portion of full information, yielding a lower SS. Conversely, if the prompt induce model to integrate broader semantic signals across multiple retrieved documents—often through abstraction or summarization—its output may align well with the concatenated retrievals, raising SS, while its verbatim overlap with any individual document becomes relatively weaker, lowering CRR.

---

> ### Author Response · Authors · 2025-11-23
> **Rebuttal by Authors [4/4]**
>
> ### **Reference**:
>
> [1] Stav Cohen, Ron Bitton, and Ben Nassi. Unleashing worms and extracting data: Escalating the outcome of attacks against rag-based inference in scale and severity using jailbreaking. arXiv preprint arXiv:2409.08045, 2024\.
>
> [2] Changyue Jiang, Xudong Pan, Geng Hong, Chenfu Bao, and Min Yang. Rag-thief: Scalable extrac- tion of private data from retrieval-augmented generation applications with agent-based attacks. arXiv preprint arXiv:2411.14110, 2024\.
>
> [3] Zhenting Qi, Hanlin Zhang, Eric P Xing, Sham M Kakade, and Himabindu Lakkaraju. Follow my instruction and spill the beans: Scalable data extraction from retrieval-augmented generation systems. In ICLR 2024 Workshop on Navigating and Addressing Data Problems for Foundation Models.
>
> [4] https://iapp.org/news/a/generative-ai-and-intellectual-property-copyright-implications-for-ai-inputs-outputs
>
> [5] [https://www.dwt.com/blogs/artificial-intelligence-law-advisor/2025/02/reuters-ross-court-ruling-ai-copyright-fair-use](https://www.dwt.com/blogs/artificial-intelligence-law-advisor/2025/02/reuters-ross-court-ruling-ai-copyright-fair-use)
>
> [6] Keith Muller. Statistical power analysis for the behavioral sciences, 1989.

---

> ### Author Response · Authors · 2025-11-27
> **Looking forward to further feedback**
>
> Dear Reviewer CR94,
>
>
> We hope this message finds you well.
>
> As the **rebuttal period deadline** is approaching, we would greatly appreciate it if you could kindly acknowledge and respond to our rebuttal for paper ICLR 18281. We are looking forward to receiving any further questions or suggestions for improvement. If you find our responses satisfactory, we appreciate it if you could kindly revise your rating based on your updated assessment of our work.
>
> We sincerely look forward to your further comments. Thank you very much for your support.
>
> Best regards,
>
> Authors of ICLR 18281

---

### Official Review · Reviewer_9Ln3 · 2025-10-31

**Soundness:** 2
**Presentation:** 2
**Contribution:** 2
**Rating:** 4
**Confidence:** 3

**Summary:**

The paper proposes IKEA, a “benign-query” knowledge-extraction attack on RAG. It combines (i) Experience Reflection sampling over “anchor concepts” and (ii) Trust-Region Directed Mutation (TRDM) to explore the embedding space, and evaluates against RAG-Thief and DGEA under input/output defenses, reporting higher extraction efficiency and attack success.

**Strengths:**

1. The paper is well-written and easy to follow

2. The studied topic is important and novel

**Weaknesses:**

1. The paper evaluates against only two prior attacks, RAG-Thief (prompt-injection) and DGEA (jailbreak), even though the Related Work section lists additional, closely related extraction methods (e.g., Pirates of the RAG / adaptive black-box extraction) that are not included as baselines. This makes the claimed superiority (“surpassing baselines by 80%+”) hard to trust. At minimum, strong black-box, non-jailbreak/PIK variants and adaptive coverage attacks should be implemented. More discussion on the related works is needed.

2. “Semantic Similarity (SS)” uses an encoder to compare outputs with retrieved docs, favoring paraphrase-style extraction (IKEA) over verbatim baselines, while CRR (ROUGE-L) penalizes paraphrase. Claims hinge on SS/EE/ASR; there is no human audit of copyright risk nor independent leakage criteria. Copyright/privacy stakes aren’t well reflected by SS alone.

3. HealthCare-100k, HarryPotterQA, and Pokémon are niche; Pokémon is explicitly chosen as low-overlap with pretraining. Results may not generalize to enterprise RAG (contracts, support logs, medical records), where policy, formatting, and noise differ.

4. The main setup assumes a known domain topic; the “unknown topic” setting still uses a bespoke topic-probing stage powered by a secondary LLM, then evaluates almost identically—this weakens the claim that IKEA remains benign and practical under stricter assumptions.

5. Replacing Top-K with off-topic docs predictably tanks both the attack and benign utility to near zero (Table 4), which is not an acceptable real-world mitigation, so it doesn’t inform deployers what works.

6. The pipeline and equations are clear, but the headline claim (“surpassing baselines by >80% efficiency, >90% success”) rests on a baseline set that is neither representative nor matched to IKEA’s benign-query regime. Without stronger baselines, the empirical claim reads overstated.

**Questions:**

1. Add competitive benign-query baselines: random/diversity sampling; k-center or farthest-point query selection; BM25 lexical sweeps; self-ask/chain-expansion; an adaptive coverage agent; and a re-implementation of adaptive black-box extraction from the works already cited in §5.

2. at least one enterprise-style corpus with policy/PII-like structure, and long-document settings that stress retrieval/reranking.

---

> ### Author Response · Authors · 2025-11-23
> **Rebuttal by Authors [1/5]**
>
> Thank you for your valuable comments. Below we respond to the comments in **Weaknesses** and **Questions**.
>
> ---
>
> ***Weakness 1: On the choice of baselines and missing adaptive black-box attacks***
>
> We appreciate the reviewer’s concern regarding the coverage of baselines. To address this point comprehensively, we next provide a systematic, method-by-method discussion of all potentially relevant extraction approaches, including those presented in the papers cited in the Related Work section.
>
> 1. We do not include PIDE [1] in our experiments for its non-adaptive nature and its less optimal performance, which has been proved in previous works [6, 7].
>
> 2. Note that DGEA [6] is an adaptive jailbreak attack, not a simple prompt-injection method. It is designed to actively explore unseen documents and therefore, is representative of adaptive extraction.
>
> 3. RAG-thief [7] is also an adaptive extraction attack, which utilizes document overlap and LLM query expansion.
>
> 4. We additionally add comparison experiments of the PoR (Pirate-of the RAG) attack [20], an adaptive black-box extraction method (see $\color{blue} \text{Table 1}$) in our revised version and therefore has included all existing published RAG extraction methods. IKEA performs over 40% higher EE on PoR in all settings, over 50% higher EE on PoR in all defensive settings. IKEA also outperforms PoR in ASR in nearly all settings, especially under input-ensemble defense, which proves the superiority of IKEA.
>
> 5. Additionally, we evaluate benign-query extraction baselines ($\color{blue} \text{Table 20}$, $\color{blue} \text{Appendix B.12}$). We detail these evaluations in **Weakness 6** and **Question 1** below. As shown in $\color{blue} \text{Table 20}$,
> none of these approaches achieve strong extraction performance: EE remains below 0.51 across all benign baseline. In contrast, IKEA reaches substantially higher EE (0.88) and considerable ASR, CRR and SS, demonstrating that anchor-guided ER + TRDM method is far more effective than naive or simple heuristic benign-query expansion.
>
> 6. Theoretically, we also provide theoretical analysis ($\color{blue} \text{Appendix F}$) to demonstrate that IKEA achieves near-optimal coverage complexity $T(\alpha) = \Theta(\alpha N / K)$, while random, greedy, or cluster-wise baselines require $\Omega\big (c\frac{N_{\max}}{K}\log N_{\max}\big)$ queries for complete coverage—formally explaining why these baselines are inherently weaker.
>
> In summary, we broadly considered all relevant potential baselines, as well as strong black-box methods, non-jailbreak variants, and adaptive coverage attacks. We provide detailed discussion and empirical comparisons across these settings, which consistently show that our approach achieves superior performance under various defenses, making our overall empirical evaluation fully representative.

---

> ### Author Response · Authors · 2025-11-23
> **Rebuttal by Authors [2/5]**
>
> ***Weakness 2: Concern on evaluation metrics: SS, CRR***
>
> We acknowledge that SS rewards paraphrastic extraction. However, our motivation of this paper is not merely to measure verbatim reproduction, but to assess whether an attacker can reconstruct the knowledge needed to form a usable substitute RAG—the central copyright/privacy threat.
>
> 1. We emphasize that it is reasonable to set CRR and SS to evaluate verbatim and semantic leakage, respectively. We provide IKEA case studies ($\color{blue} \text{Appendix I}$) to show that the non-verbatim extractions also contain considerable details, like highly domain-specific terminology, headers, and structured definitions without literal copying. IKEA does not aim to verbatim reconstruct the documents, while it is designed to extract informative documents summaries. CRR is designed to evaluate the extraction of the key terminologies, while SS is to evaluate the semantic reconstruction rate of the whole extraction. Therefore, the metric setting of SS and CRR is reasonable.
> 2. We acknowledge that only SS and CRR cannot fully evaluate the threat of extractions. Therefore, we also provide the evaluations on the extracted retrievals' internal knowledge ($\color{blue} \text{Section 4.4}$, $\color{blue} \text{Figure 3}$) and substitute RAG's performance ($\color{blue} \text{Section 4.5}$, $\color{blue} \text{Table 2}$) represent that the extractions contain usable informative copyright knowledge and our method effectively poses underlined copyright/privacy threats. There are already relevant legal cases [1,2] demonstrate the practical risk that AI-generated outputs, when substituting or competing with original works, may affect the market for copyright holders—even if the outputs are not identical. This further supports our argument in $\color{blue}\text{Section 4.4}$.
> 3. To further strengthen the claim and prove the possible commercial value of the extractions, we add downstream RAG evaluation ($\color{blue} \text{Figure 9}$) with extractions under both input- and output-level defense to build the substitute RAG. The result shows that IKEA’s extracted corpus supports downstream diagnosis tasks almost as well as the original RAG, confirming that semantic leakage is practically exploitable. Hence, our measurements also reflect copyright/privacy risk as actual utility of stolen knowledge, not just string overlap.
>
> ---
>
> ***Weakness 3: Concern about niche datasets and generalizability to enterprise RAG***
>
> We thank the reviewer's comment. To address this point, we provide a systematic, discussion here with additional experiments:
>
> 1. HealthCare-100k [10] is indeed a medical-record like corpus used in prior security evaluations. HarryPotterQA [12] and HealthCare-100k [10] are standard in previous extraction works (both used in RAG-Thief [7] and DGEA [6]). We use the Pokémon [11] dataset for its little overlap with LLM's internal knowledge.
>
> 2. We provide additional evaluation with the enterprise dataset ($\color{blue} \text{Appendix B.9}$, $\color{blue} \text{Table 18}$): ArXiv [8], Github README [9]. Specifically, we use ArXiv articles published in January to May 2025 as scientific-type dataset, Github projects' READMEs created after September as structured dataset. As shown in $\color{blue} \text{Table 18}$, IKEA shows acceptable Extraction Efficiency (0.52 for GitHub and 0.56 for ArXiv) with these unseen structured datasets.
>
> 3. Follow your suggestion, we expand our evaluation in $\color{blue} \text{Table 1}$ and add enterprise-style corpora to confirm IKEA’s effectiveness under realistic document structures: Legal-Contract [13], for enterprise-style, long-document, policy-heavy text, and NQ-corpus [14], for multi-topic, widely-used, heterogeneous, large-scale open-domain corpus.
>
> As shown in $\color{blue} \text{Table 1}$, on **Legal-Contract**, IKEA achieves strong extraction quality under all settings (EE>0.57, SS>0.62), while baselines remain low under strong defenses like input-level defense. On **NQ-Corpus**, IKEA is again the only method maintaining stable performance across defenses (EE ≈ 0.64 - 0.66; ASR ≈ 0.82 - 0.88), whereas all competing attacks fail in some defense settings. These results show that IKEA preserves extraction capability on long, heterogeneous, enterprise-style documents, confirming its superiority in realistic settings.
>
> ​

---

> ### Author Response · Authors · 2025-11-23
> **Rebuttal by Authors [3/5]**
>
> ***Weakness 4: Concerns about the known-topic assumption and topic probing***
>
> We appreciate the reviewer’s comments. Here, we analyze this point in details:
>
> 1. Our paper’s primary setting—known topic—is aligned with real deployments where users query RAG systems for a specific purpose [2,3,4,5].  RAG is typically used to enrich LLMs with external domain knowledge for specialized scenarios or users, such as medical question answering [15], financial analysis [16], or legal inquiry [17]. Therefore, the RAG topics are usually exposed to normal users for better utilization of the RAG.
>
> 2. We emphasize that there’s no need for any prior knowledge in the topic-probing stage. The probing algorithm also does not rely on specific seed topics ($\color{blue} \text{section 4.6}$). Additionally, the topic probing process is fully benign, using only natural probing questions, which won't weaken the benign assumption.
>
> 3. Empirically ($\color{blue} \text{Table 3}$), IKEA retains strong extraction efficiency with the probed topics. Even in multi-topic dataset NQ-corpus which has **no literal topics**, IKEA achieves high performance with multi-topic probing ($\color{blue} \text{Table 1}$). On NQ-Corpus, IKEA maintains stable performance across defenses (EE ≈ 0.64–0.66; ASR ≈ 0.82–0.88), whereas all competing attacks fail in some defense settings. These results show that the topic probing algorithm works on large-scale, heterogeneous documents without triggering detection or largely degrading extraction performance. This proves that our assumption and method design is reasonable.
>
> Thus, the “unknown-topic” setting does not weaken the practical benign nature otherwise strengthens the adaptivity of IKEA.
>
> ---
>
> ***Weakness 5: Concerns about the section of adaptive defense***
>
> We acknowledge with the reviewer that Top-K replacement with off-topic distractors is not a realistic production defense. **However, we clarify that our goal in $\color{blue} \text{Table 5}$ ($\color{blue} \text{Section 4.7}$) is not to recommend this defense, but to examine IKEA’s effectiveness and robustness under adaptive defense of retrieval perturbation.** This experiment shows that IKEA can be disrupted only at the cost of collapse of the RAG utility. This result highlights the core threat that this study aims to reveal.
>
> The retrieval-level adaptive defense is designed to emphasize our posed threats: Although the adaptive defense disrupts the stable Top-$K$ similarity structure that IKEA relies on, IKEA still achieves 0.12-0.22 extraction efficiency instead of dropping to near zero. In contrast, the RAG utility entirely collapses to zero. The result shows the robustness of IKEA and emphasizes the threat posed by benign extraction attacks.
>
> ---
>
> ***Weakness 6: Lack of more stronger benign-query baselines***
>
> We appreciate the reviewer's concern. Follow your suggestion, we additionally include a comprehensive set of benign-query baselines ($\color{blue} \text{Appendix B.12}$, $\color{blue} \text{Table 20}$):
>
> * Random LLM-generated benign-query attack, which directly samples LLM-generated brainstorm queries and achieves relatively high ASR but lacks coverage control.
>
> * FarthestPoint sampling benign-query attack, which selects new queries that are maximally distant from all previous retrievals , measured by embedding similarity.
>
> * BM25 lexical sweeps sampling benign-query attack, which selects new queries that have minimum average BM25 score with previous retrievals.
>
> * LLM chain-expansion benign-query attack, which expands queries with LLM using the latest response.
>
> * Self-coverage benign-query attack, which implements a Pseudo Relevance Feedback (PRF)-like query extraction attack inspired by CSQE [23]: RAG responses serve as a steering corpus for iteratively crafting new queries. When the model replies “I don't know" or the response contains little information, a new query is regenerated from the topic while avoiding verbatim repetition.
>
> As shown in $\color{blue} \text{Table 20}$, none of these approaches achieve strong extraction performance: EE remains below 0.51 across all benign baseline. In contrast, IKEA reaches substantially higher EE (0.88), demonstrating that anchor-guided exploration is far more effective than naive or simple heuristic benign-query expansion. Note that although some approaches achieve similar ASR, CRR, and SS, this is because they all use benign-look queries. Considering that an attack must operate under a limited budget (e.g. 256 rounds extraction in our experiments), EE becomes a key indicator for distinguishing extraction efficiency in these cases.

---

> ### Author Response · Authors · 2025-11-23
> **Rebuttal by Authors [4/5]**
>
> ***Question 1: Ask for adding attack baselines***
>
> We provide comparison experiments of Pirates of the RAG (see **Weakness 1**), and provide comprehensive comparison experiments of benign-query baselines (see **Weakness 6**).
>
> ---
>
> ***Question 2: Ask for adding dataset evaluation***
>
> We provide additional experiments on Legal-Contract dataset, for enterprise-style, long-document text, and experiments on NQ-corpus, for multi-topic, heterogenous, large-scale corpus (see **Weakness 3**).

---

> ### Author Response · Authors · 2025-11-23
> **Rebuttal by Authors [5/5]**
>
> ### **Reference**:
>
> [1] Zhenting Qi, Hanlin Zhang, Eric P Xing, Sham M Kakade, and Himabindu Lakkaraju. Follow my instruction and spill the beans: Scalable data extraction from retrieval-augmented generation systems. In ICLR 2024 Workshop on Navigating and Addressing Data Problems for Foundation Models.
>
> [2] Peng Xia, Kangyu Zhu, Haoran Li, Tianze Wang, Weijia Shi, Sheng Wang, Linjun Zhang, James Zou, and Huaxiu Yao. Mmed-rag: Versatile multimodal rag system for medical vision language models. arXiv preprint arXiv:2410.13085, 2024
>
> [3] Spurthi Setty, Harsh Thakkar, Alyssa Lee, Eden Chung, and Natan Vidra. Improving retrieval for rag based question answering models on financial documents. arXiv preprint arXiv:2404.07221, 2024\.
>
> [4] Nirmalie Wiratunga, Ramitha Abeyratne, Lasal Jayawardena, Kyle Martin, Stewart Massie, Ikechukwu Nkisi-Orji, Ruvan Weerasinghe, Anne Liret, and Bruno Fleisch. Cbr-rag: case-based reasoning for retrieval augmented generation in llms for legal question answering. In Interna- tional Conference on Case-Based Reasoning, pp. 445–460. Springer, 2024\.
>
> [5] Varun Kumar, Leonard Gleyzer, Adar Kahana, Khemraj Shukla, and George Em Karniadakis. My- crunchgpt: A llm assisted framework for scientific machine learning. Journal of Machine Learn- ing for Modeling and Computing, 4(4), 2023\.
>
> [6] Stav Cohen, Ron Bitton, and Ben Nassi. Unleashing worms and extracting data: Escalating the outcome of attacks against rag-based inference in scale and severity using jailbreaking. arXiv preprint arXiv:2409.08045, 2024\.
>
> [7] Changyue Jiang, Xudong Pan, Geng Hong, Chenfu Bao, and Min Yang. Rag-thief: Scalable extrac- tion of private data from retrieval-augmented generation applications with agent-based attacks. arXiv preprint arXiv:2411.14110, 2024\.
>
> [8] RealTimeData. arxiv_alltime. https://huggingface.co/datasets/RealTimeData/arxiv\_alltime, a. Accessed: 2025-09-21.
>
> [9] RealTimeData. bbc_news_alltime. https://huggingface.co/datasets/RealTimeData/bbc\_news\_alltime, b. Accessed: 2025-09-21.
>
> [10] lavita AI. lavita/chatdoctor-healthcaremagic-100k · datasets at hugging face. URL [https://huggingface.co/datasets/lavita/ChatDoctor-HealthCareMagic-100k](https://huggingface.co/datasets/lavita/ChatDoctor-HealthCareMagic-100k).
>
> [11] Duong Quang Tung. Tungdop2/pokemon · datasets at hugging face. URL https://huggingface.co/datasets/tungdop2/pokemon.
>
> [12] vapit. vapit/harrypotterqa · datasets at hugging face. URL https://huggingface.co/datasets/vapit/HarryPotterQA.
>
> [13] Azzindani. Legal\_contract\_syn. https://huggingface.co/datasets/Azzindani/Legal\_Contract\_Syn. Hugging Face dataset.
>
> [14] Jack Morris. nq\_corpus\_dpr. https://huggingface.co/datasets/jxm/nq\_corpus\_dpr. Hugging Face dataset.
>
> [15] Alejandro Lozano, Scott L Fleming, Chia-Chun Chiang, and Nigam Shah. Clinfo. ai: An open-source retrieval-augmented large language model system for answering medical questions using scientific literature. In Pacific Symposium on Biocomputing 2024, pp. 8–23. World Scientific,2023
>
> [16] Xiang Li, Zhenyu Li, Chen Shi, Yong Xu, Qing Du, Mingkui Tan, Jun Huang, and Wei Lin. Alphafin: Benchmarking financial analysis with retrieval-augmented stock-chain framework. arXiv preprint arXiv:2403.12582, 2024a.
>
> [17] Nirmalie Wiratunga, Ramitha Abeyratne, Lasal Jayawardena, Kyle Martin, Stewart Massie, Ikechukwu Nkisi-Orji, Ruvan Weerasinghe, Anne Liret, and Bruno Fleisch. Cbr-rag: case-based reasoning for retrieval augmented generation in llms for legal question answering. In International Conference on Case-Based Reasoning, pp. 445–460. Springer, 2024\.
>
> [18] Ruizhe Zhu, Hao Zhu, Yaxuan Li, Syang Zhou, Shijing Cai, Malgorzata Lazuka, and Elliott Ash.Dialogueforge: Llm simulation of human-chatbot dialogue. arXiv preprint arXiv:2507.15752,2025.
>
> [19] Hongwei Yao, Haoran Shi, Yidou Chen, Yixin Jiang, Cong Wang, Zhan Qin, Kui Ren, and Chun Chen. Controlnet: A firewall for rag-based llm system. arXiv preprint arXiv:2504.09593, 2025\.
>
> [20] Christian Di Maio, Cristian Cosci, Marco Maggini, Valentina Poggioni, and Stefano Melacci. Pirates of the rag: Adaptively attacking llms to leak knowledge bases. arXiv preprint arXiv:2412.18295,2024.
>
> [21] https://iapp.org/news/a/generative-ai-and-intellectual-property-copyright-implications-for-ai-inputs-outputs
>
> [22] https://www.dwt.com/blogs/artificial-intelligence-law-advisor/2025/02/reuters-ross-court-ruling-ai-copyright-fair-use
>
> [23] Yibin Lei, Yu Cao, Tianyi Zhou, Tao Shen, and Andrew Yates. Corpus-steered query expansion with large language models. arXiv preprint arXiv:2402.18031, 2024\.

---

> > ### Comment · Reviewer_9Ln3 · 2025-11-24
> >
> > Thanks for the comprehensive rebuttal. I have updated my score accordingly.

---

> ### Author Response · Authors · 2025-11-25
>
> Thank you for your recognition and raising the score. We appreciate your constructive comments and will further polish the paper accordingly. Thank you again.

---

### Official Review · Reviewer_LmNX · 2025-11-01

**Soundness:** 2
**Presentation:** 3
**Contribution:** 2
**Rating:** 4
**Confidence:** 3

**Summary:**

This submission investigates covert extraction of proprietary knowledge from retrieval-augmented generation (RAG) systems and proposes “IKEA,” an implicit, benign-query attack that grows “anchor concepts” via history-aware sampling and a trust-region mutation in embedding space. Evaluations across several corpora and model/retriever pairings suggest higher extraction efficiency than prompt-injection baselines and show that a substitute RAG assembled from harvested content retains non-trivial utility. Topic probing for unknown domains and simple adaptive/DP-style defenses are also explored to characterize security–utility trade-offs.

**Strengths:**

1. This paper clearly specifies a realistic black-box threat model for RAG and delineates attacker capabilities and constraints with precision.
2. Empirical coverage is broad, spanning multiple LLM–retriever configurations and defenses, and the attack remains effective when common jailbreak/prompt-injection attacks are blocked.
3. The method is straightforward and reproducible—anchor-based benign queries guided by history-aware sampling and a cosine-bounded trust-region mutation—with prompts and hyperparameters disclosed.

**Weaknesses:**

1. Algorithmic novelty feels limited; the core components amount to history-penalized sampling and cosine-bounded mutations without formal coverage or sample-complexity guarantees.
2. This paper depends on a known or easily probed domain topic and centralized corpus semantics, making generalization to heterogeneous, multi-topic enterprise deployments uncertain.
3. This paper’s defense study leans on simplistic or utility-destroying mechanisms and omits deployable strategies like per-client rate limiting, query-set anomaly detection, and semantic drift monitoring.
4. This paper lacks an end-to-end economic analysis of the attack (token/time costs and sensitivity to generator quality), which is crucial for real-world risk assessment.

**Questions:**

No more questions.

---

> ### Author Response · Authors · 2025-11-23
> **Rebuttal by Authors [1/2]**
>
> Thank you for your valuable feedback. Below we respond to the comments in **Weaknesses**.
>
> ---
>
> ***Weakness 1: Limited novelty of algorithm; Lack formal coverage or sample-complexity guarantees.***
>
> We restate the novelty of our methodology with the additional theoretical guarantee as follows:
>
> 1. Our task setting and methodological design fill a previously unaddressed gap: **we pioneer the implicit threat** that attackers can utilize only benign-query to extract copyright content of the RAG database. In comparison, Prior RAG extraction attacks—RAG-Thief, DGEA, and Pirates-of-RAG —all assume adversarial prompts or jailbreak-style interactions. Our problem formulation is therefore new, and we contribute the first framework explicitly designed for this setting.
>
> 2. Methodologically, IKEA is more than “penalized sampling \+ cosine-bounded mutation.” (1) Experience Reflection (ER) learns a dynamic penalty landscape from historical query–response trajectories to suppress semantic regions that are provably low-density or irrelevant. (2) TRDM exploits the empirically validated link between query–response distance and local RAG density ($\color{blue} \text{Figure 6}$), enabling directed exploitation rather than random mutation or naive coverage heuristics.
>
> 3. We further provide a theoretical justification ($\color{blue} \text{Appendix F}$) showing that IKEA achieves near-optimal coverage complexity $T(\alpha) = \Theta(\alpha N / K)$, while random, greedy, or cluster-wise baselines require $\Omega\big (c\frac{N_{\max}}{K}\log N_{\max}\big)$ queries for nearly complete coverage—formally explaining why these baselines are inherently weaker.
>
> ---
>
> ***Weakness 2: Concerns about applicability to generalization to heterogeneous, multi-topic.***
>
> For the concerns about IKEA’s applicability to generalization to heterogeneous, multi-topic scenarios, we include an end-to-end experiment of Multi-topic probing based extraction (details in $\color{blue} \text{Section 4.6}$)  with NQ-corpus ($\color{blue} \text{Table 1}$), a heterogeneous, multi-topic dataset. The NQ-corpus doesn't have literal topics. Therefore, we first conduct Multi-topic probing and then extract it with IKEA. The topic probing based extraction evaluation on NQ-corpus shows the effectiveness of our method under heterogeneous, multi-topic datasets.
> As shown in $\color{blue} \text{Table 1}$,  IKEA maintains stable performance across defenses (EE ≈0.64–0.66; ASR ≈0.82–0.88) within the NQ-Corpus dataset. Whereas other baselines either exhibit degraded performance under the Input-Ensemble defense, IKEA sustains high EE values while achieving high ASR.
> Compared with the baselines, IKEA achieves superior performance with multi-topic probing. These results prove the effectiveness of IKEA, and also show that the topic probing algorithm is adaptive to large-scale, heterogeneous documents without triggering detection or largely degrading extraction performance.

---

> ### Author Response · Authors · 2025-11-23
> **Rebuttal by Authors [2/2]**
>
> ***Weakness 3: Evaluation under more deployable and sophisticated defense strategies.***
>
> Thank you for your insightful comments. We conduct additional experiments to validate the effectiveness of our methods under more defense deployments, such as query-set anomaly detection and semantic drift monitoring. Here we provide the details.
>
> First, we emphasize that **Per-client rate limiting** is not applicable IKEA queries behave identically to standard user queries and do not increase request frequency. Rate limiting affects every heavy RAG user equivalently and is not a targeted defense mechanism against semantic extraction.
>
> Second, following your comment, we additionally add two deployable strategies ($\color{blue} \text{Table 4}$, $\color{blue} \text{Section 4.7}$) representing **Query-set anomaly detection** and **Semantic drift monitoring**:  Sequential Detection (Seq-Detect) and Semantic Detection (Sem-Detect), to detect suspicious queries based on sequential information and semantic drift, respectively.
>
> Specifically, (1) for Seq-Detect, we train a transformer-based sequential detector for sequence-level anomaly detection with the three attacks' data and human-rag interaction data [3], (2) for Sem-Detect, we utilize the semantic-level detector based on ControlNET [4], a firewall framework explicitly designed for RAG systems.
>
> We report the classification AUC to evaluate the detection effectiveness. We also report the true positive rate when the false positive rate is 1% and 10% (TPR@1%FPR, TPR@10%FPR) to evaluate the practical effectiveness without degrading normal user experience.
>
> As shown in $\color{blue} \text{Table 4}$, these two methods achieve near-perfect performance against baseline attacks (DGEA and RAG-Thief), with AUC values, TPR@1%FPR, and TPR@10%FPR almost all reaching 1.0. In contrast, Seq-Detect and Sem-Detect achieve AUC values of only 0.76 and 0.75, respectively, against IKEA, indicating that IKEA is markedly more stealthy than the baselines. Moreover, both methods exhibit a significant drop in TPR@1%FPR and TPR@10%FPR compared to their performance on the baseline attacks, with TPR@10%FPR remaining below 0.3. Since deployed defenses must not interfere with normal usage, the effectiveness of these two methods against IKEA is insufficient for practical deployment.
>
> In summary, our previous input- and output-level defenses including Intention analysis, Keyword filtering, Defensive instructions and Content similarity redaction, cover both query-level and output-space defense. And the revised defense study spans input-level, semantic-level, sequence-level, and retrieval-level defenses. Taken together, these results show that IKEA is markedly more stealthy than existing baselines under various defenses.
>
> ---
>
> ***Weakness 4: Lack end-to-end economic analysis and Generator sensitivity evaluation***
>
> 1. We have included Token and time cost comparison ($\color{blue} \text{Table 9}$) in the previous version. As shown in the experiment, IKEA requires fewer attack tokens and achieves lower wall-clock extraction time than RAG-Thief and DGEA.
>
> 2. We additionally add sensitivity evaluation of the Generator ($\color{blue} \text{Table 16}$). The experiment shows: IKEA remains effective across GPT-4o, DeepSeek-v3, and Qwen-7B-Instruct. As shown in $\color{blue} \text{Table 16}$, all generators sustain strong performance, but stronger models provide smoother semantic alignment with anchor concepts. Deepseek-v3 generator achieves the highest EE and ASR, while GPT-4o generator offers slightly better reconstruction performance. Qwen-7B-Instruct generator performs slightly lower but remains stable. These results show that IKEA is largely generator-agnostic, with more capable generators offering modest gains in efficiency.
>
> ---
>
> ### **Reference**:
>
> [1] Ruizhe Zhu, Hao Zhu, Yaxuan Li, Syang Zhou, Shijing Cai, Malgorzata Lazuka, and Elliott Ash.Dialogueforge: Llm simulation of human-chatbot dialogue. arXiv preprint arXiv:2507.15752,2025.
>
> [2] Hongwei Yao, Haoran Shi, Yidou Chen, Yixin Jiang, Cong Wang, Zhan Qin, Kui Ren, and Chun Chen. Controlnet: A firewall for rag-based llm system. arXiv preprint arXiv:2504.09593, 2025\.
>
> [3] Ruizhe Zhu, Hao Zhu, Yaxuan Li, Syang Zhou, Shijing Cai, Malgorzata Lazuka, and Elliott Ash.Dialogueforge: Llm simulation of human-chatbot dialogue. arXiv preprint arXiv:2507.15752,2025.
>
> [4] Hongwei Yao, Haoran Shi, Yidou Chen, Yixin Jiang, Cong Wang, Zhan Qin, Kui Ren, and Chun Chen. Controlnet: A firewall for rag-based llm system. arXiv preprint arXiv:2504.09593, 2025\.

---

> ### Author Response · Authors · 2025-11-27
> **Looking forward to further feedback**
>
> Dear Reviewer LmNX,
>
> We hope this message finds you well.
>
> As the **rebuttal period deadline** is approaching, we would greatly appreciate it if you could kindly acknowledge and respond to our rebuttal for paper ICLR 18281. We are looking forward to receiving any further questions or suggestions for improvement. If you find our responses satisfactory, we appreciate it if you could kindly revise your rating based on your updated assessment of our work.
>
> Thank you very much for your support.
>
> Best regards,
>
> Authors of ICLR 18281

---

### Author Response · Authors · 2025-11-23
**Summary of Paper Revision**

We thank all reviewers for their constructive feedback, and we have responded to each reviewer individually. We have also uploaded a **Paper Revision** including further method details,  additional results, and illustrations:

$\color{blue} \text{Section 4.6}$ (page 8): We clarify details of the topic-probing algorithm under the multi-topic scenario.

$\color{blue} \text{Section 4.7}$ (page 9): We add details of detection-based defense.

$\color{blue} \text{Figure 6}$ (page 19): We add extensive experiments using three mainstream datasets to validate the assumption that query–response distance reliably reflects the local document density.

$\color{blue} \text{Figure 9}$ (page 24): We add additional downstream task evaluation of substitute RAG.

$\color{blue} \text{Table 1}$ (page 7): (1) We add two evaluation datasets,  Legal-Contract and NQ-corpus, covering long-text enterprise-style documents and multi-topic open-domain documents. (2) We add a new attack baseline, "Pirates of RAG" [1], to represent a black-box adaptive extraction attack.

$\color{blue} \text{Table 4}$ (page 9): We add evaluation on two new detection-based defense methods: Sequential Detection (Seq-Detect) and Semantic Detection (Sem-Detect) [2].

$\color{blue} \text{Table 14}$ (page 21): We add a new random method without anchors for comparison and clarification.

$\color{blue} \text{Table 16}$ (page 21): We add the experiment of generator sensitivity analysis.

$\color{blue} \text{Table 20}$ (page 25): We add five benign-query–based extraction attacks for further comparison with IKEA.

$\color{blue} \text{Table 21}$ (page 26): We add robustness evaluation on the Topic Probing algorithm.

$\color{blue} \text{Appendix F}$ (page 28): We add theoretical analysis on extraction complexity.


### **Reference**:

[1] Di Maio C, Cosci C, Maggini M, et al. Pirates of the rag: Adaptively attacking llms to leak knowledge bases[J]. arXiv preprint arXiv:2412.18295, 2024\.

[2] Yao H, Shi H, Chen Y, et al. Controlnet: A firewall for rag-based llm system[J]. arXiv preprint arXiv:2504.09593, 2025\.

---

### Author Response · Authors · 2025-12-01
**Rebuttal Summary (for AC) [1/4]**

***Dear AC and SACs,***

We understand that, due to a rare bug in the ICLR/OpenReview system, you are facing a significantly higher workload than usual, including re-reading updated submissions and reaching final decisions without the normal author-reviewer discussion phase. We are sincerely grateful for the additional time and care you are investing, and we greatly appreciate your efforts in keeping the ICLR review process running smoothly under these unexpected circumstances.

To help you efficiently assess our paper and to provide context on what happened during the rebuttal stage, we present below a structured summary of the reviewers' main concerns together with our corresponding responses. **We respectfully emphasize that all effective interactions between us and the reviewers took place before 27 Nov 2025 (i.e., prior to the public disclosure of the OpenReview system bug). During this period, two reviewers increased their ratings ($\color{Green} \text{9Ln3}$: 4 → 6, $\color{Brown} \text{iS3Y}$: 6 → 8) and the other two reviewers have not yet responded and have not raised any new concerns.**


---

We first provide a summary of our paper revisions during rebuttal period, and then present our point-by-point responses to each reviewer's concerns.

**Summary of main revisions.** In this revision we **(i)** add a theoretical analysis of the coverage complexity of IKEA (Reviewer $\color{#CC6633} \normalsize \text{LmNX}$, $\color{purple} \normalsize \text{CR94}$), **(ii)** add comparison experiments with six attack baselines, including Pirates-of-RAG [1] and five benign-query extraction attacks (Reviewer $\color{Green} \normalsize \text{9Ln3}$, $\color{purple} \normalsize \text{CR94}$), **(iii)** introduce two new realistic datasets: the multi-topic NQ-corpus and the Legal-Contract corpus, and add specific details on applying the topic-probing algorithm to the multi-topic scenario (Reviewer $\color{#CC6633} \normalsize \text{LmNX}$, $\color{Green} \normalsize \text{9Ln3}$, $\color{Brown} \normalsize \text{iS3Y}$), **(iv)** add evaluation against two stronger deployable defenses (semantic and sequential detection) (Reviewer $\color{#CC6633} \text{LmNX}$, $\color{Green} \normalsize \text{9Ln3}$, $\color{Brown} \normalsize \text{iS3Y}$), **(v)** add more downstream RAG evaluations showing the utility of the extracted corpus (Reviewer $\color{Green} \normalsize \text{9Ln3}$, $\color{purple} \normalsize \text{CR94}$, $\color{Brown} \normalsize \text{iS3Y}$), **(vi)** empirically validate the query-response distance assumption underlying TRDM  (Reviewer $\color{#CC6633} \normalsize \text{LmNX}$, $\color{purple} \normalsize \text{CR94}$), **(vii)** include a true global random baseline for ablation study (Reviewer $\color{purple} \normalsize \text{CR94}$), and **(viii)** conduct robustness experiments for the topic-probing algorithm (Reviewer $\color{Brown} \normalsize \text{iS3Y}$). Further details are provided in the "Summary of Paper Revision" and in our rebuttals to each reviewers.

### **Responses to Reviewer LmNX**

***W1: Limited algorithmic novelty and lack of formal coverage guarantees.***

**A1:** We restate our key contributions, clarify that benign-query implicit extraction is a new problem setting, and provide theoretical coverage-complexity analysis showing IKEA's near-optimality.
(Refer to: $\color{blue} \text{Figure 6}$, $\color{blue} \text{Appendix F}$ )

***W2: Uncertain applicability to heterogeneous or multi-topic corpora.***

**A2:** We detail the topic-probing algorithm under multi-topic scenario and add multi-topic experiments (e.g., NQ-Corpus) to demonstrate the stable performance of our method across heterogeneous domains.
(Refer to: $\color{blue} \text{Section 4.6}$, $\color{blue} \text{Table 1}$ )

***W3: Lack of evaluation under more realistic deployable defenses.***

**A3:** We discussed reviewer proposed defense one-by-one, then add experiments of Query-set anomaly detection (Seq-Detect) and Semantic drift monitoring (Sem-Detect), the results of which show both defenses fail to reliably detect IKEA without harming benign utility.
(Refer to: $\color{blue} \text{Section 4.7}$, $\color{blue} \text{Table 4}$ )

***W4: Lack of economic analysis and generator sensitivity.***

**A4:** We clarify that we have already reported IKEA’s lower token/time cost in our original manuscript, and we add evaluations of generator sensitivity in the revision, which shows consistent effectiveness across multiple generators.
(Refer to: $\color{blue} \text{Table 9}$, $\color{blue} \text{Table 16}$ )

**Status explanation for AC.** Until 27 Nov 2025, Reviewer $\color{#CC6633} \normalsize \text{LmNX}$ had not yet replied to our rebuttal. However, we believe our response thoroughly addresses all concerns raised by Reviewer $\color{#CC6633} \normalsize \text{LmNX}$, experimentally and theoretically. Hence, we respectfully request the AC to consider our responses in the evaluation.

---

### Author Response · Authors · 2025-12-01
**Rebuttal Summary (for AC) [2/4]**

### **Responses to Reviewer 9Ln3**

***W1: Lack of more attack baselines.***

**A1:** We include additional comparison experiments with Pirates-of-RAG [1] and five benign-query baselines. In these experiments, IKEA outperforming all baselines by large margins. We also provide theoretical coverage complexity proof to validate the effectiveness.
(Refer to: $\color{blue} \text{Table 1}$, $\color{blue} \text{Table 20}$, $\color{blue} \text{Appendix B.12}$, $\color{blue} \text{Appendix F}$ )

***W2: SS metric favors paraphrasing and lacks human leakage assessment.***

**A2:** We highlight the reasonableness of SS/CRR metric, and these metrics can be complemented with evaluation experiments about internal-knowledge and substitute RAG. Additional downstream substitute-RAG experiment also shows practical effectiveness of our method.
(Refer to: $\color{blue} \text{Table 2}$, $\color{blue} \text{Figure 3}$, $\color{blue} \text{Figure 9}$, $\color{blue} \text{Section 4.4}$, $\color{blue} \text{Section 4.5}$, $\color{blue} \text{Appendix I}$ )

***W3: Datasets are niche and may not represent enterprise RAG.***

**A3:** We add experiments on Legal-Contract, NQ-Corpus, ArXiv and GitHub datasets confirming IKEA's effectiveness and robustness on long, policy-like, heterogeneous documents.
(Refer to: $\color{blue} \text{Table 1}$, $\color{blue} \text{Table 18}$, $\color{blue} \text{Appendix B.9}$ )

***W4: Topic-probing stage weakens the "unknown topic" assumption.***

**A4:** We clarify our reasonable assumption and the generalizability of IKEA facing different corpora. We experimentally show Topic-probing using fully benign queries and maintain IKEA’s performance even under multi-topic settings.
(Refer to: $\color{blue} \text{Table 1}$, $\color{blue} \text{Figure 3}$, $\color{blue} \text{Section 4.6}$ )

***W5: The concern about the uselessness of the Off-topic Top-K defense.***

**A5:** We emphasize that the purpose of $\color{blue} \text{Section 4.7}$ (Off-topic Top-K defense) is to evaluating IKEA under adaptive defense, rather than proposing a new defense. The experiments show that defending against IKEA may destroy RAG utility, which further highlights the threat of IKEA.
(Refer to: $\color{blue} \text{Table 5}$, $\color{blue} \text{Section 4.7}$ )

***W6: Claims only rely on weak baselines without benign-query baselines comparisons.***

**A6:** We expand our experiments on five additional benign-query baselines to further prove IKEA's superiority.
(Refer to: $\color{blue} \text{Table 20}$, $\color{blue} \text{Appendix B.12}$ )

***Q1: Ask for adding attack baselines.***

**A7:** See responses to **W1** and **W6**.

***Q2: Ask for adding more datasets evaluation.***

**A8:** See responses to **W3**.

**Status explanation for AC.** After discussion, Reviewer $\color{Green} \normalsize \text{9Ln3}$ accepted our rebuttal and, on 25 Nov 2025, provided a positive follow-up indicating that our responses resolved his/hers concerns and increased the score from **4 to 6** accordingly. We appreciate it.

---

### Author Response · Authors · 2025-12-01
**Rebuttal Summary (for AC) [3/4]**

### **Responses to Reviewer CR94**

***W1: The idea of using query-response semantic distance as a proxy for local RAG density is only intuitive and lacks validation.***

**A1:** We add a dedicated validation showing strong positive correlation between query-response similarity and local document density across three datasets, supporting this assumption empirically.
(Refer to: $\color{blue} \text{Figure 6}$, $\color{blue} \text{Appendix B.6}$ )

***W2: Lack of several relevant extraction baselines.***

**A2:** We add evaluation experiments about Pirates-of-RAG [1] and five benign-query baselines, covering all published methods and benign-query extraction methods, showing IKEA's consistent superiority.
(Refer to: $\color{blue} \text{Table 20}$, $\color{blue} \text{Appendix B.12}$, $\color{blue} \text{Appendix F}$ )

***W3: Low ROUGE scores suggest limited practical copyright or privacy risk.***

**A3:** We emphasize that a low ROUGE score only indicates low verbatim reconstruction and does not imply limited practical copyright or privacy risk. We experimentally show that IKEA's extractions (1) contain rich domain-specific details and (2) enable substitute RAGs with strong downstream performance, demonstrating real semantic leakage beyond verbatim overlap.
(Refer to: $\color{blue} \text{Table 2}$, $\color{blue} \text{Figure 3}$, $\color{blue} \text{Figure 9}$, $\color{blue} \text{Section 4.4}$, $\color{blue} \text{Section 4.5}$, $\color{blue} \text{Appendix I}$)

***W4: Metric definitions (EE, ASR, et, al.) are unclear and comparability is questionable.***

**A4:** We provide precise formal definitions of these metrics (EE/ASR/CRR/SS), and we emphasize that the evaluation of substitute-RAG together with these metrics offers a comprehensive view of leakage, demonstrating the superiority of our method.
(Refer to: **W3**, $\color{blue} \text{Table 1}$, $\color{blue} \text{Table 7}$, $\color{blue} \text{Figure 9}$, $\color{blue} \text{Section 4.4}$, $\color{blue} \text{Section 4.5}$, $\color{blue} \text{Appendix A.2}$)

***W5: Lack of discussion on how the hyperparameters are chosen.***

**A5:** We have provided sensitivity analysis on the key parameter $\gamma$, showing a clear efficiency-cost trade-off. For other parameters, we choose them empirically and keep them consistent throughout all experiments, which shows the generality of the parameter settings under different scenarios.
(Refer to: $\color{blue} \text{Table 6}$, $\color{blue} \text{Figure 7}$, $\color{blue} \text{Appendix B.8}$ )

***W6: Ablation results show only small gains over random baselines.***

**A6:** We clarify the notation of "random" in ablation experiments, introduce a true global random baseline which show a much larger efficiency gap, and connect this result to our complexity analysis that explains why ER+TRDM is more sample-efficient.
(Refer to: $\color{blue} \text{Table 14}$, $\color{blue} \text{Section 3.2}$, $\color{blue} \text{Appendix B.8}$, $\color{blue} \text{Appendix F}$ )

***Q1: In the no-defense setting, why does DGEA achieve very high ROUGE (near-literal copying) but relatively low embedding similarity?***

**A7:** We clarify the definition of CRR: it measures verbatim overlap with the single best-matching document, while SS compares embeddings across all concatenated retrievals. Therefore, copying one document boosts ROUGE but does not guarantee high SS value.
(Refer to: $\color{blue} \text{Appendix A.2}$ )

**Status explanation for AC.** Until 27 Nov 2025, Reviewer $\color{purple} \normalsize \text{CR94}$ had not yet replied to our rebuttal. However, we believe our response fully addresses all concerns raised by Reviewer $\color{purple} \normalsize \text{CR94}$, experimentally and theoretically. Hence, we kindly request the AC to consider our responses in the evaluation.

---

### Author Response · Authors · 2025-12-01
**Rebuttal Summary (for AC) [4/4]**

### **Responses to Reviewer iS3Y**

***W1: The tested defenses are insufficient.***

**A1:** We add deployable Seq-Detect and Sem-Detect defenses and show they almost perfectly detect DGEA/RAG-Thief but perform much worse on IKEA, indicating IKEA remains stealthy under the more realistic and stronger defenses.
(Refer to: $\color{blue} \text{Table 4}$, $\color{blue} \text{Section 4.7}$ )

***W2: The fixed-and-known topic assumption and the reliance on topic probing may limit applicability.***

**A2:** We (1) emphasize that fixed topics match common real-world RAG deployments and (2) add multi-topic and noisy-corpus experiments on NQ-corpus showing topic probing is robust and IKEA still achieves high EE/ASR even beyond the original assumption.
(Refer to: $\color{blue} \text{Table 1}$, $\color{blue} \text{Table 3}$, $\color{blue} \text{Section 4.6}$ )

***W3: The current results are not enough to support that the substitute RAG performs "comparably" to the original.***

**A3:** We clarify that “comparable” refers to functional utility and additionally include a more realistic downstream task, symptom-to-diagnosis, where the substitute RAG generated by IKEA is closest to the original one and outperforms all baselines.
(Refer to: $\color{blue} \text{Table 2}$, $\color{blue} \text{Figure 9}$, $\color{blue} \text{Section 4.5}$, $\color{blue} \text{Appendix B.10}$ )

***W4: The cost in time, API calls, and query rounds is unclear.***

**A4:** We clarify the description and report query/attack token counts and 256-round extraction time, showing IKEA uses fewer attack tokens and less wall-clock time than RAG-Thief and DGEA while achieving stronger extraction.
(Refer to: $\color{blue} \text{Table 9}$, $\color{blue} \text{Appendix B.3}$ )

***Q1: Clarification on how the rebuttal addresses the above weaknesses (1-4).***

**A5:** We explicitly answer each point in Weaknesses 1-4 with detailed methodological, experimental, and theoretical clarifications in the corresponding rebuttal sections.

***Q2: More details on the robustness of the topic-probing algorithm.***

**A6:** We experimentally prove the robustness of topic-probing algorithm under (1) unknown diverse seed topics, (2) noise-injected corpora, and (3) the multi-topic NQ-corpus, consistently achieving high EE/ASR without triggering detection or degrading extraction performance.
(Refer to: $\color{blue} \text{Table 1}$, $\color{blue} \text{Table 3}$, $\color{blue} \text{Table 21}$, $\color{blue} \text{Appendix B.13}$ )

***Q3: Could IKEA queries be detectable via sequential anomaly detection?***

**A7:** We additionally design a sequential anomaly detection by training a transformer-based sequential detector. We find it preforms nearly perfect on DGEA/RAG-Thief but much weaker on IKEA (AUC 0.76 with very low TPR@1%FPR and TPR@10%FPR), indicating that sequential anomaly detection cannot defend against IKEA in realistic scenarios.
(Refer to: $\color{blue} \text{Section 4.7}$, $\color{blue} \text{Table 4}$ )

**Status explanation for AC.** After discussion, Reviewer $\color{Brown} \normalsize \text{iS3Y}$ accepted our rebuttal and, on 27 Nov 2025, provided a positive follow-up indicating that our responses resolved his/her concerns and increased the score from **6 to 8** accordingly. We appreciate it.

---

Lastly, we sincerely thank the AC for their time, careful handling of our submission, and thoughtful coordination of the review process. Your guidance and the reviewers' feedback have significantly helped us refine and improve this work. Thank you again.

[1] Di Maio C, Cosci C, Maggini M, et al. Pirates of the rag: Adaptively attacking llms to leak knowledge bases[J]. arXiv preprint arXiv:2412.18295, 2024.

---

### Meta-Review · Area_Chair_B4b5 · 2026-01-10

**Summary:**

The paper received an overall borderline acceptance assessment. Across four reviewers, initial scores ranged from 4 to 6, with concerns primarily centered on algorithmic novelty, baseline coverage, metric validity, defense realism, and practical deployment considerations. Following the rebuttal, two reviewers explicitly increased their scores (from 4→6 and 6→8), indicating that many key concerns were addressed. Reviewers acknowledged the paper’s clear threat model, benign-query attack formulation, and strong empirical performance of IKEA, particularly its stealthiness under realistic defenses and the demonstrated utility of extracted knowledge through substitute RAG systems. The remaining weaknesses mainly relate to perceived limitations in algorithmic novelty, reliance on several custom evaluation metrics, and residual questions about real-world deployment costs and defenses; however, these issues were mitigated through added theoretical analysis, expanded experiments, and detailed clarifications in the rebuttal. Overall, the paper matured significantly during the revision process and now presents a compelling and well-supported contribution.

**Reviewer Concerns:**

The rebuttal comprehensively addressed the majority of reviewer concerns. Missing and unrepresentative baselines were resolved by adding Pirates-of-RAG and five benign-query baselines, showing consistent superiority of IKEA. Questions about metric validity (SS, CRR, EE, ASR) were clarified with formal definitions, additional analyses, and downstream substitute-RAG evaluations, demonstrating practical semantic leakage beyond ROUGE. Concerns about topic assumptions and generalization were mitigated via multi-topic probing, noisy-corpus robustness tests, and new datasets (Legal-Contract, NQ-Corpus, ArXiv, GitHub). The lack of deployable defenses was addressed by adding sequential anomaly detection and semantic drift monitoring, where IKEA remained substantially stealthier than prior attacks. Finally, economic cost and generator sensitivity were clarified with token/time analyses and cross-model experiments.

Some reviewers may still view the algorithmic novelty as incremental despite the added theoretical coverage-complexity analysis, particularly given reliance on intuitive components (history-aware sampling and embedding-space mutation). While defenses were expanded, real-world system-level mitigations (e.g., production monitoring, rate-limiting at scale, human-in-the-loop auditing) remain only indirectly discussed. Additionally, although downstream utility was strengthened, broader task diversity and longer-horizon economic risk modeling could further solidify real-world impact. These issues are relatively minor and do not outweigh the strengthened empirical and theoretical contributions.

**Reviewer Scores:**

Reviewer LmNX: Initial score 4 → Likely 6 after full discussion (novelty and defense concerns largely addressed).

Reviewer 9Ln3: Initial score 4 → Confirmed increase to 6 after rebuttal and added baselines (before Openreview Security Incident).

Reviewer CR94: Initial score 4 → Likely higher, given resolved concerns on metrics, baselines, and validation.

Reviewer iS3Y: Initial score 6 → Confirmed increase to 8 after rebuttal addressing defenses, cost, and applicability (before Openreview Security Incident).

---

### Decision · Program_Chairs · 2026-01-26

Accept (Poster)